# Deletion of Notch3 Impairs Contractility of Renal Resistance Vessels Due to Deficient Ca^2+^ Entry

**DOI:** 10.3390/ijms232416068

**Published:** 2022-12-16

**Authors:** Frank Helle, Michael Hultström, Panagiotis Kavvadas, Bjarne Iversen, Christos E. Chadjichristos, Christos Chatziantoniou

**Affiliations:** 1INSERM UMR S 1155, Hôpital Tenon, 75020 Paris, France; 2Faculty of Medicine, Sorbonne University, 75013 Paris, France; 3Renal Research Group, Institute of Medicine, University of Bergen, 5007 Bergen, Norway; 4Department of Medicine, Haukeland University Hospital, 5021 Bergen, Norway; 5Department of Biomedicine, University of Bergen, 5007 Bergen, Norway; 6Integrative Physiology, Department of Medical Cell Biology, Uppsala University, 75105 Uppsala, Sweden; 7Anaesthesiology and Intensive Care Medicine, Department of Surgical Sciences, Uppsala University, 75105 Uppsala, Sweden

**Keywords:** Notch3, renal haemodynamics, Ca^2+^ mobilization, renal resistance vessels, vasoreactivity

## Abstract

Notch3 plays an important role in the differentiation and development of vascular smooth muscle cells. Mice lacking Notch3 show deficient renal autoregulation. The aim of the study was to investigate the mechanisms involved in the Notch3-mediated control of renal vascular response. To this end, renal resistance vessels (afferent arterioles) were isolated from Notch3^−/−^ and wild-type littermates (WT) and stimulated with angiotensin II (ANG II). Contractions and intracellular Ca^2+^ concentrations were blunted in Notch3^−/−^ vessels. ANG II responses in precapillary muscle arterioles were similar between the WT and Notch3^−/−^ mice, suggesting a focal action of Notch3 in renal vasculature. Abolishing stored Ca^2+^ with thapsigargin reduced Ca^2+^ responses in the renal vessels of the two strains, signifying intact intracellular Ca^2+^ mobilization in Notch3^−/−^. EGTA (Ca^2+^ chelating agent), nifedipine (L-type channel-blocker), or mibefradil (T-type channel-blocker) strongly reduced contraction and Ca^2+^ responses in WT mice but had no effect in Notch3^−/−^ mice, indicating defective Ca^2+^ entry. Notch3^−/−^ vessels responded normally to KCl-induced depolarization, which activates L-type channels directly. Differential transcriptomic analysis showed a major down-regulation of *Cacna1h* gene expression, coding for the α_1H_ subunit of the T-type Ca^2+^ channel, in Notch3^−/−^ vessels. In conclusion, renal resistance vessels from Notch3^−/−^ mice display altered vascular reactivity to ANG II due to deficient Ca^2+^-entry. Consequently, Notch3 is essential for proper excitation–contraction coupling and vascular-tone regulation in the kidney.

## 1. Introduction

Notch is a family of membrane receptors participating in cell growth, differentiation, and apoptosis [1,2]. Out of the four isoforms (Notch1–4), the Notch3 receptor is specifically expressed in smooth vasculature [3] and has been shown to be crucial for the maturation of small arteries [4]. Notch3 expression seems to be functionally important only in certain vascular beds, such as in tail [5] and cerebral arteries [4], but not in the carotid artery [5] or skin [6].

In humans, a mutation in Notch3 causes the disorder cerebral autosomal-dominant arteriopathy with subcortical infarcts and leukoencephalopathy (CADASIL), which is an inherited small-vessel disease that causes ischemic strokes, mental disorders, and premature death [7,8]. In a genetic mouse model designed to mimic CADASIL, vascular smooth muscle cell degeneration was shown to precede the pathological changes [9,10], suggesting that vascular dysregulation is likely to play a primary role in the pathogenesis of the disease.

The importance of renal vascular reactivity in controlling blood pressure led our group to study renal haemodynamics in mice lacking the Notch3 receptor [6]. Compared to wild-type littermates, Notch3^−/−^ mice displayed a significantly blunted renal vascular response after systemic bolus injections of vasoactive agents, such as norepinephrine, angiotensin II (ANG II), bradykinin, or prostacyclin, consistent with deficient renal autoregulation. Furthermore, chronic infusion of ANG II was associated with increased mortality and severe renal damage in Notch3^−/−^ mice [6]. These data indicate that Notch3 is essential for controlling renal vascular tone. However, in these previous studies, we did not elucidate the underlying cellular mechanism(s) involved in this dysfunction. In the present study, we investigated the contractile properties and Ca^2+^ responses of renal resistance vessels’ freshly isolated afferent arterioles (AAs) from Notch3^−/−^ mice.

Despite the expression of Ca^2+^ channels in many segments of the renal vascular tree [11], we focused on the AA because the afferent segment is of major importance when translating the ANG II signal to a vascular response [12] and is therefore most relevant for the changes observed in renal blood flow and renal vascular resistance in our earlier study [6]. Furthermore, defects of the afferent segment of the renal cortex are most compatible with nephropathy previously observed in some CADASIL patients [13,14].

Therefore, AAs isolated from Notch3^−/−^ and wild-type mice were challenged with ANG II. The use of calcium channels or intracellular Ca^2+^ inhibitors indicated that the absence of Notch3 compromised entry of extracellular Ca^2+^. The underlying mechanism of this defect was related to a significant downregulation of the α_1H_ subunit belonging to the T-type Ca^2+^ channel. It thus appears that Notch3 regulates renal vascular tone by controlling the expression and function of specific Ca^2+^ channels.

## 2. Results

### 2.1. Notch3 Deficiency Affects Vascular Contractility in a Tissue-Specific Manner

To test whether Notch3 affects vascular responses in a similar way in different tissues, we compared contractility responses to ANG II in a dose-dependent manner between abdominal muscle precapillary arterioles and AAs. Baseline diameters were similar in wild-type and Notch3^−/−^ AAs (24.3 ± 2.1 µm vs. 24.2 ± 1.6 µm, respectively) and muscle resistance vessels (14.0 ± 2.1 vs. 14.0 ± 2.2 µm, respectively). As expected, the mean diameter of renal AAs isolated from wild-type mice decreased in a dose-dependent manner, reaching 60% of baseline at 10^−7^ M ANG II (Figure 1A). In contrast, renal vessels isolated from Notch3^−/−^ mice showed little responsiveness to ANG II (barely a 10% decrease at 10^−7^ M ANG II, *p* < 0.01, Figure 1A). This strain difference was not observed when abdominal microvessels were challenged with the same doses of ANG II (60 ± 4% vs. 57 ± 7%, respectively, Figure 1B). These results suggest that the expression of Notch3 affects vessel contractility in a tissue-specific manner.

### 2.2. Notch3^−/−^ Mice Display Decreased Levels of the T-Type Ca^2+^ Channel Subunit α_1H_ in the Renal Cortex

To elucidate the mechanism underlying the observed defective contractile response to acute administration of ANG II in AAs, we performed transcriptomic analysis on renal cortical slices from Notch3^−/−^ and wild-type mice (Table A1). We focused on 47 genes well-known to control smooth muscle contractility. We found that the most striking difference between the two strains was the downregulation of Cacna1h, a gene coding the α_1H_ subunit of the T-type Ca^2+^ channel (Figure 2, Table A1).

### 2.3. Blunted Contractile Responses of AAs from Notch3^−/−^ Mice Are Associated with Decreased Intracellular Ca^2+^ (Ca^2+^_i_) Levels

Renal vessels from Notch3^−/−^ and wild-type mice displayed similar baseline values of Ca^2+^_i_ (fura-2 ratio: 0.81 ± 0.03 vs. 0.83 ± 0.02, respectively). However, when vessels were challenged with ANG II, contraction and Ca^2+^_i_ responses were weaker in AAs isolated from Notch3^−/−^ kidneys (*p* < 0.01, Figure 3A,E). This result is consistent with the vessel diameter dose-responses shown in Figure 1.

### 2.4. Extracellular Entry of Ca^2+^ Is Compromised in AAs from Notch3^−/−^ Mice

When Ca^2+^ was removed from the medium with EGTA, contractions and Ca^2+^ responses to ANG II were reduced in vessels from wild-type mice, but were unchanged in vessels from Notch3^−/−^ mice (Figure 3B,F). As a result, the strain difference seen under basal conditions disappeared. Similarly, when Ca^2+^ was prevented from entering the cell through L- (Figure 3C,G) or T-type (Figure 3D,H) Ca^2+^ channels (inhibited with nifedipine and mibefradil, respectively), contractions were significantly blunted in vessels from wild-type mice. In contrast, the addition of these inhibitors did not affect the Ca^2+^ responses or contractility of Notch3^−/−^ AAs. Again, the strain difference in the contractility and Ca^2+^ responses disappeared, suggesting that the reason for the improper response of the Notch3^−/−^ vessels to ANG II was due to dysfunction in extracellular Ca^2+^ entry.

### 2.5. Intracellular Ca^2+^ Mobilization Is Normal in AAs from Notch3^−/−^ Mice

To examine whether compromised Ca^2+^_i_ mobilization also contributed to blunted contractions, we treated AAs with the Ca^2+^ ATPase inhibitor thapsigargin, which abolishes intracellular calcium stores (Figure 4A,B). Thapsigargin treatment significantly reduced the peak Ca^2+^ response in both strains (*p* < 0.001), indicating that intracellular Ca^2+^ mobilization was equally efficient in the two strains.

### 2.6. AAs from Notch3^−/−^ Mice Depolarize Normally

Since previous results indicated defects in calcium entry, we tested the function of L-type Ca^2+^-entry in vessels from Notch3^−/−^ mice using KCl-induced depolarization. Depolarization produced identical contractions and Ca^2+^ responses in both strains (Figure 5A,C), which were abolished by nifedipine blockade (Figure 5B,D).

## 3. Discussion

Previous studies from our lab have shown that deletion of Notch3 in mice results in deficient in vivo renal responses to vasoactive agents. In the present study, our objective was to identify the mechanism(s) underlying the abnormal renal vascular response in mice lacking Notch3. To this end, we examined ANG II-induced contractility and Ca^2+^_i_ concentrations in renal resistance vessels isolated from Notch3^−/−^ mice. We found that the renal vasculature of Notch3^−/−^ mice does not respond normally to ANG II, possibly due to a deficient entry of Ca^2+^.

Notch3^−/−^ mice showed no differences in terms of body or kidney weight compared to wild-type littermates. Despite a thinner vessel wall of uneven thickness, as reported before in Notch3^−/−^ [4,6], the two strains exhibited similar lumen diameter and baseline cytosolic Ca^2+^ concentrations. When challenged with ANG II, however, it was apparent that contractions and Ca^2+^ responses to ANG II were much weaker in renal vessels of Notch3^−/−^ mice (Figure 1A and Figure 3A,E). It is important to note that the defective response was specific to renal vessels since pre-capillary vessels of other tissues responded normally (Figure 1B).

This defective response can be due to either impaired mobilization of Ca^2+^_i_ and/or Ca^2+^ entry. We tested the hypothesis of Ca^2+^ mobilization by using thapsigargin, a potent inhibitor of sarco/endoplasmic reticulum Ca^2+^-ATPase. Thapsigargin addition in the medium completely blocked Ca^2+^ spikes in the afferent arterioles of WTs, as previously reported in vascular smooth muscle cells of resistance arteries [15]. An important decrease was also observed in afferent arterioles from Notch3^−/−^ mice, suggesting that Ca^2+^ mobilization is not deregulated in the renal vessels of these mice. The fact that the observed decrease was bigger in the WTs’ arterioles was due to higher levels of Ca^2+^ responses in a normal medium.

Removal of extracellular Ca^2+^ from the medium using EGTA strongly reduced contractility and Ca^2+^ concentration in normal renal vessels, in agreement with previous studies [16]. In sharp contrast, EGTA had a negligible effect on the responses of Notch3^−/−^ vessels (Figure 3B,F). As result, the strain difference disappeared, and the contractility and Ca^2+^ concentrations became similar. These results strongly suggest that the defective response of Notch3^−/−^ vessels is due to dysfunction of Ca^2+^ entry. Renal afferent arterioles display two types of Ca^2+^ channels: L-type and T-type [11]. Nifedipine, an L-type calcium blocker mimicked the effect of EGTA (reduced response in WT vessels, no effect in Notch3^−/−^ vessels), suggesting dysfunctional L-type channels. However, the expression of L-type channels did not differ between the vessels of the Notch3^−/−^ and WT (Table A1), and the results with KCl-induced depolarization (Figure 5) indicate that L-type channels are functioning correctly in Notch3^−/−^.

The T-type Ca^2+^ channel is primarily expressed in preglomerular AAs [11]. Mibefradil, a T-type calcium blocker, also mimicked the effect of EGTA (reduced response in WT vessels; no effect in Notch3^−/−^ vessels). In this case however, we observed a major decrease in the expression of the α_1H_ subunit of the T-type channel (*Cacna1h*) in Notch3^−/−^ renal vessels (Table A1). In fact, *Cacna1h* was the most-affected gene compared to 46 other genes related to contractile function. We can thus hypothesize that the T-type Ca^2+^ channel is dependent on the Notch3 pathway for its expression on renal vessels. Alternatively, since a lack of Notch3 decreases smooth muscle actin expression and deforms the cytoskeleton and structure of the vascular wall (Table A1 [5,6,17]), it is possible that the expression of T-type Ca^2+^ on cell membranes was affected in Notch3^−/−^ vessels. The low expression of this channel can explain, at least partly, the deficiency in the vascular response in the preglomerular vessels of Notch3^−/−^ mice.

The question, however, remains: Why was Ca^2+^ entry from L-type channels also compromised? One possibility is that the decreased response of Notch3^−/−^ afferent arterioles to ANG II is not due to a dysfunction of Ca^2+^ entry from its channels but is caused by an abnormality somewhere in the signalling from the ANG II receptor to the Ca^2+^ channels. This hypothesis is not supported by our previous study testing renal vascular responses in Notch3^−/−^ mice [6]. In the previous study, we found deficient responses of Notch3^−/−^ kidneys to other vasoconstrictors (such as norepinephrine or thromboxane) and to mechanical stress as well (which is independent of any kind of G-protein-receptor signalling pathway). This deficient response is extended to vasodilators, such as prostacyclin or bradykinin as well, showing that the deficient vasoconstriction is not due to a compensatory activation of a vasodilatory signalling.

Another hypothesis is that L- and T-type channels are interacting or cross-communicating, and a proper Ca^2+^ entry requires the appropriate functioning of both. The T-type channels are involved in excitability [18,19,20,21] and therefore impact the overall responsiveness of the vessel. Related to this role, it was reported that L- and T-type Ca^2+^ depend on each other to initiate contraction. In rabbit AAs, L-type Ca^2+^ entry is reliant on functional T-type Ca^2+^ channels [11]. Conversely, a blockade of the T-type Ca^2+^ channel has no effect on mice lacking the L-type channel [11]. In line with these findings, L- and T-type channel-blockers have no additive effects on dilation in rat juxtaglomerular AAs, regardless of the order in which they are administered [22]. Using KCl depolarization and nifedipine, other investigators concluded that contractility and Ca^+2^ entry in afferent arterioles depend mainly on L-type channels with little contribution from T-type channels [23,24,25]. It is possible, however, that the low-current, transient T-type calcium flow happens but is unable to activate the high-conductance L-type channels because it cannot displace the nifedipine blockade. As a consequence, KCl-induced vasoconstriction is inhibited in the presence of nifedipine. Finally, it appears that that Ca^2+^ entry and mobilization in afferent arterioles make up a very complex mechanism involving interactions between L- or T- type, transient receptor potential, and K^+^ and/or Cl^−^ channels as well [26]. It is possible that an interruption of this complex interplay at one step could affect the whole pathway of Ca^2+^ responses and contractility.

Alternatively, Ca^2+^ entry can be compromised in Notch3^−/−^ renal vessels because the lack of the *Notch3* gene is accompanied by defects in vascular smooth muscle cell maturation [4,6]. Supporting this hypothesis, it was observed that Notch3-dependent postnatal arterial maturation did not correlate with the expression of the Notch3 ligands Jagged1 and Delta4 [4]. Instead, maturation of arterial vascular smooth muscle cells closely parallels the increase in arterial blood pressure during foetal development [27]. Based on these data, it was proposed that the transduction of pressure and shear stress from arteriolar blood flow to the vessel wall, necessary for arterial maturation of the actin cytoskeleton, is dependent on Notch3 signalling [4]. In agreement with this notion, we observed that the vascular wall of Notch3^−/−^ afferent arterioles displayed an uneven thickness due to an incomplete maturation of vascular smooth muscle cells [6].

Whatever the nature of this interaction, our data show that the afferent arterioles of Notch3^−/−^ mice depend mainly on intracellular Ca^2+^_i_ stores in order to respond to cardiovascular stress and vasoconstrictor stimuli. During control conditions, this appears to be adequate for maintaining vasoregulatory functions. During pathological conditions, however, such as those induced by prolonged exposure to exogenous ANG II [6], requiring extracellular Ca2^+^ influx into the cells, mice lacking Notch3 fail to regulate renal haemodynamics, causing severe renal dysfunction and injury.

In our previous study of renal haemodynamics in Notch3^−/−^ mice, we found a blunted and likely insufficient increase in RVR compared to wild-type ones after injecting 0.5 ng ANG II into the bloodstream. In contrast, blood pressure, indicating total peripheral resistance, increased equally in the two strains [6]. Consequently, while the kidney seems unable to generate a normal pressure response, the majority of resistance vessels in the Notch3^−/−^ mouse must have a normal contractile response to ANG II. Consistent with this notion, we found a normal response to ANG II in muscle resistance vessels from Notch3^−/−^, and we have previously obtained similar data from skin resistance vessels [6].

Other investigators found an almost passive increase in cerebral blood flow after ANG II injections in Notch3^−/−^, but not wild-type, mice, while the blood pressure responses were similar [4]. Interestingly, another subunit of the T-type channel, α_1G_, was shown to be downregulated several-fold in Notch3^−/−^ cerebral vascular smooth muscle cells (SI Appendix Table 2A [28]). Taken together, these and our studies suggest that Notch3^−/−^ vascular dysfunction in the brain and kidney may have molecular and cellular mechanistic similarities. Although different in some respects, the renal and cerebral vasculature share a vital physiological mechanism, the autoregulation of haemodynamics.

The list of genes in Table A1 shows that, in addition to *Cacna1h* (encoding the T-type Ca^2+^ channel), several genes affecting vessel contractility were also downregulated in the absence of Notch3. Examples from this list include connexin 43 (gap-junctions), transient receptor potential C6 (receptor-operated Ca^2+^ entry [29]), integrin-linked kinase (Ca^2+^-independent contraction [30]), and rhoA (Notch3-dependent mechanotransduction [5]). Thus, it appears that additional signalling pathways controlling renal vasoreactivity are affected in Notch3^−/−^ mice. The present finding of defective Ca^2+^ entry, however, is likely the foremost mechanism affecting the ANG II response in the renal preglomerular vessels. We cannot exclude the possibility that other signalling pathways may have contributed to a lesser degree to this dysfunction, but if so, their impact was below what we can detect with the present model.

## 4. Materials and Methods

### 4.1. Animals

Experiments were performed on mice lacking expression of the *Notch3* gene (Notch3^−/−^) on a C57Bl6/J background, as described elsewhere [6]. Wild-type littermates were used as controls. A total of 62 WT and Notch3^−/−^ mice, weighing 18–25 g, were bred at the animal facility at the Institute of Biomedicine at the University of Bergen, fed ordinary mouse pellets, and had free access to tap water.

Because the objective of the study was to understand the deficient myogenic response in the kidney that we have observed previously [6], and since myogenic response depends on afferent and not on efferent arterioles, we isolated afferent arterioles from Notch3^−/−^mice, and we compared them with afferent arterioles from their WT littermates. The analytical description for the isolation of preglomerular vessels, measurements of vessel diameters, and fura-2 intracellular Ca^2+^ (Ca^2+^_i_) responses to ANG II and KCl and chemicals are provided in the Section A.2.

### 4.2. Isolation of Preglomerular Vessels

Afferent preglomerular arterioles and precapillary muscle arterioles were isolated using an agarose-infusion/enzyme-treatment technique adapted from Loutzenhiser and Loutzenhiser [28]. In short, agarose-infused kidneys were sliced and digested with enzymes for 1 h. Free-floating intact vessels with a tensile core of agarose (Figure A1) were harvested and attached to a perfusion chamber before experiments.

### 4.3. Measurement of Vessels Diameter and Ca^2+^_i_ Responses

Isolated vessels were stimulated with ANG II (10^−7^ M or dose–response curves) or depolarised with KCl (50 mM) to study the Notch3 effects on contractility and Ca^2+^ transients, (see Section A.2). In short, mean diameter using digital image analysis and Ca^2+^ ratio using fura-2 340/380 nm emission was recorded at 37° C. AAs from both sexes were used after pilot studies showed no gender differences on contractility. Nifedipine (10^−7^ M), mibefradil (10^−6^ M), and thapsigargin (10^−7^ M) were applied to the vessel bath 15 min before the recordings.

### 4.4. mRNA Extraction and Expression Analysis Using RT-PCR

Slices of the renal cortex were stabilized in RNA-later (Qiagen, Valencia, CA, USA) and frozen until use. A quantity of 15–20 mg of tissue was used to prepare the total RNA using rNeasy (Qiagen, Valencia, CA, USA). RNA was quantified using a NanoDrop ND-1000 spectrophotometer (NanoDrop Technologies, Rockland, DE, USA). Reverse transcription was performed using the RT core-kit (Eurogentec, Seraing, Belgium) with 200 ng RNA. Real-time PCR was performed on an ABI-prism 7900-HT sequence-detection system at the Norwegian Microarray Consortium in Bergen using custom-made microfluidic cards: Taqman Low-Density Array (Applied Biosystems, Carlsbad, CA, USA). A total of 47 genes known to be involved in the contractile mechanism, Notch- or calcium-signalling, and 18s were selected for RT-PCR (Table A1). Gene expressions were calculated as delta-delta-CT using 18s RNA as standard with the RT-Manager program (Applied Biosystems, Carlsbad, CA, USA).

### 4.5. Statistical Methods

Vessel diameter and Ca^2+^_i_ responses with EGTA, nifedipine, and mibefradil vs. normal medium were analysed using ANOVA followed by the student–Newman–Keuls post hoc test in SigmaStat 3.1. Diameter and Ca^2+^_i_ responses in Notch3^−/−^ vs. wild-type and thapsigargin Ca^2+^_i_ responses vs. normal medium were analysed using student’s *t*-test in SigmaStat 3.1. Gene expression was analysed using student’s *t*-test and Bonferroni’s correction. The genes were ordered after significance (Table A1), and the difference between the groups was expressed as Log2 fold-change (Log2FC). *p* < 0.05 was considered statistically significant. All values are expressed as means ± SEM.

## 5. Conclusions

The present study is, to our knowledge, the first to identify deficient Ca^2+^ entry as a major vasoactive signalling pathway dependent on Notch3 for proper function, which results in a serious defect of the calcium handling in the renal resistance vessels of mice lacking Notch3^−/−^ expression. In AAs lacking Notch3, ANG II stimulation initiated Ca^2+^ mobilization, but that was not followed by Ca^2+^ entry. These data explain our previous observation that Notch3^−/−^ kidneys are unable to cope with acute or prolonged ANG II infusion, and they also provide clues for the vascular function of Notch3 in other vascular beds, most notably the brain.

## Figures and Tables

**Figure 1 ijms-23-16068-f001:**
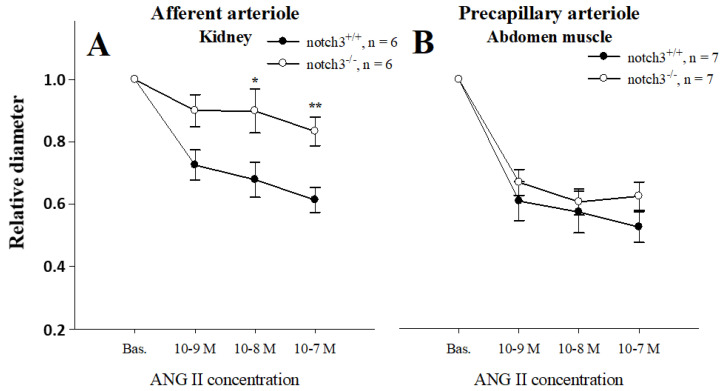
Lumen diameter dose-responses to angiotensin II (ANG II) in isolated afferent arterioles (AAs) revealed reduced contractile ability in renal vessels lacking Notch3 expression (**A**). In contrast, precapillary arterioles from abdominal muscle displayed similar contractions to ANG II (**B**). *, ** *p* < 0.05 and *p* < 0.01, respectively, vs. wild-type.

**Figure 2 ijms-23-16068-f002:**
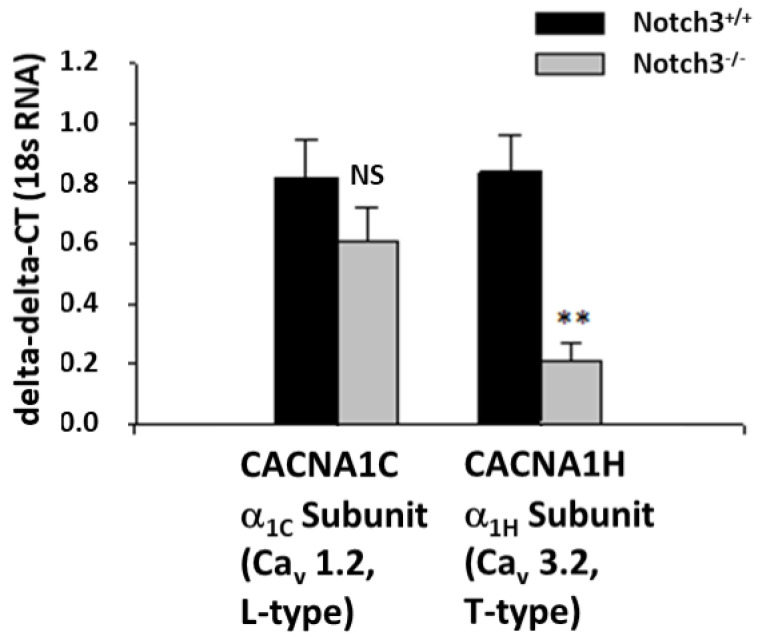
Expression of *Cacna1c* coding the α_1C_ subunit of the L-type Ca^2+^ channel did not statistically differ in renal cortex from kidneys of wild-type and Notch3^−/−^ mice. In contrast, *Cacna1h* coding the α_1H_ subunit of the T-type Ca^2+^ channel was reduced 4-fold in Notch3^−/−^ mice compared to wild-type mice (excerpt from Table A1, ** *p* = 0.014 vs. Notch^+/+^).

**Figure 3 ijms-23-16068-f003:**
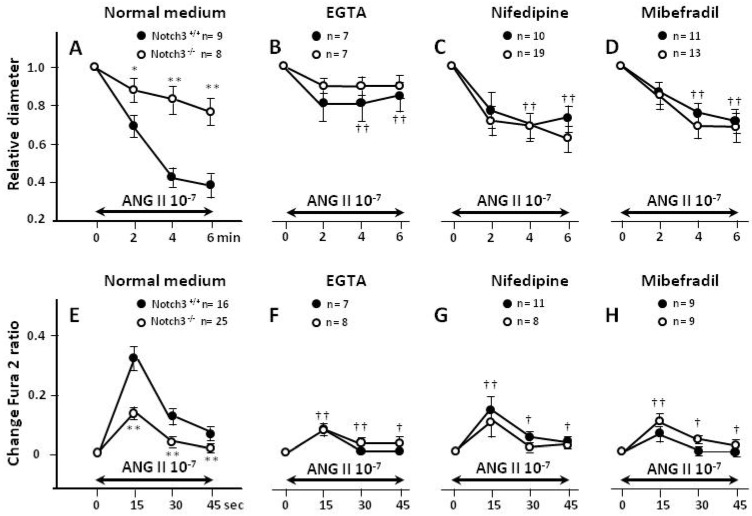
Changes in vessel diameter and Ca^2+^_i_ responses to 10^−7^ M ANG II were measured in fura-2-loaded vessels freshly isolated from wild-type and Notch3^−/−^ mice. In normal medium, both the diameter (**A**) and the Ca^2+^_i_ response (**E**) were reduced in AAs lacking Notch3 expression. This strain difference in contractile and Ca^2+^_i_ responses disappeared when Ca^2+^ was removed from the medium with EGTA (**B**,**F**), when L-type Ca^2+^ channels (Ca_v_1.2) were blocked with nifedipine (**C**,**G**), or when T-type Ca^2+^ channels (Ca_v_3.2) were blocked with mibefradil (**D**,**H**). It is noteworthy that the effect of these agents, which essentially target Ca^2+^ entry, is negligible on renal-resistant vessels from Notch3^−/−^ mice. *, ** *p* < 0.05 and *p* < 0.01, respectively, vs. WT. †, †† *p* < 0.05 and *p* < 0.01, respectively, vs. corresponding response in normal medium.

**Figure 4 ijms-23-16068-f004:**
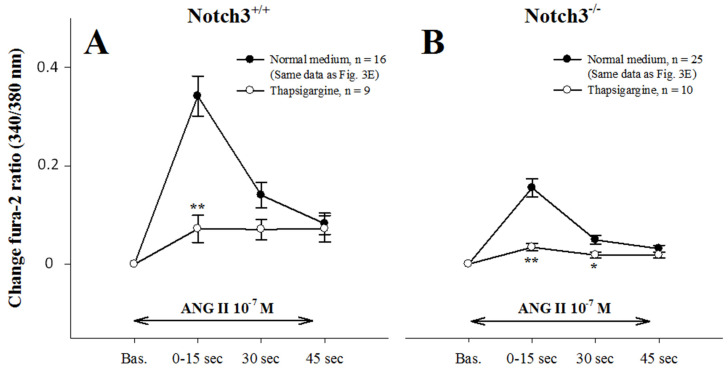
Ca^2+^ responses to 10^−7^ M ANG II, before and after administration of the Ca^2+^ ATPase inhibitor thapsigargin in wild-type (**A**) and Notch3^−/−^ (**B**) vessels. Thapsigargin prevented Ca^2+^ mobilization from intracellular stores and reduced the peak Ca^2+^ response in both strains. * and ** *p*< 0.05 and *p* < 0.001, respectively, vs. corresponding response in normal medium.

**Figure 5 ijms-23-16068-f005:**
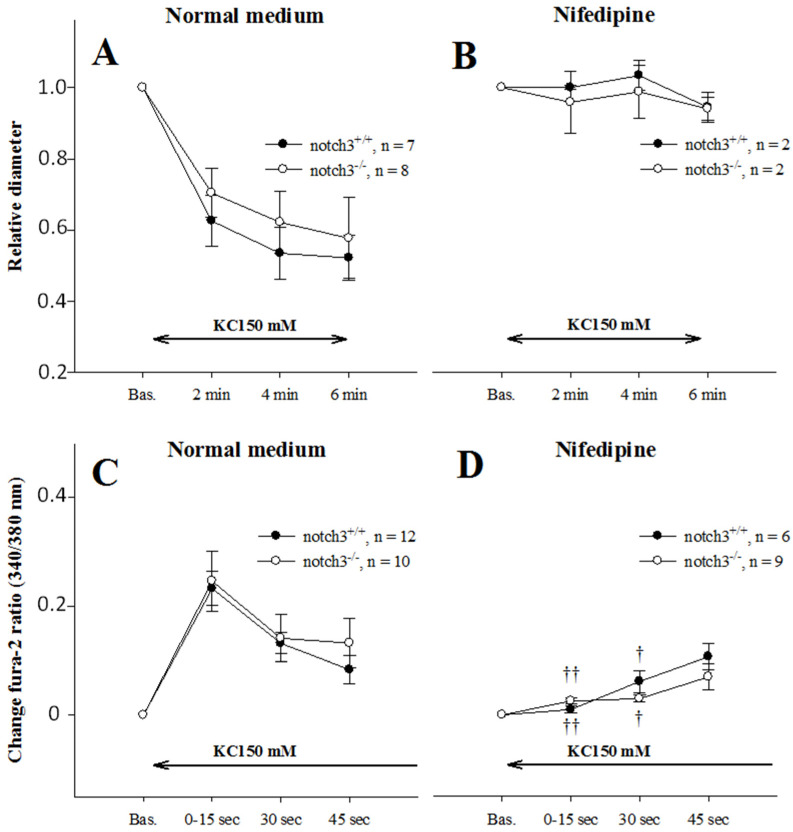
Diameter and Ca^2+^ responses after depolarization with 50 mM KCl in normal and fura-2-loaded afferent arterioles (AAs) from mice, with or without Notch3 expression. K^+^ depolarization induced almost equal diameter (**A**) and Ca^2+^ responses (**C**) in the two strains. To test specificity, vessels were depolarized again with the L-type Ca^2+^ channel-blocker nifedipine in the bath. Results showed almost abolished diameter (**B**) and Ca^2+^ responses (**D**), confirming that the L-type Ca^2+^ channel was the active isoform in both strains, †, †† *p* < 0.05 and *p* < 0.01.

## Data Availability

Not applicable.

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
