# Peer review of "Deletion of Notch3 Impairs Contractility of Renal Resistance Vessels Due to Deficient Ca2+ Entry"

_ijms, 2022, doi:10.3390/ijms232416068_

Round 1

Reviewer 1 Report

In their manuscript entitled Deletion of Notch3 impairs contractility of renal resistance vessels through a T-type Ca2+ channel-dependent mechanism, Helle et al present more data adding to the observation the group published 11 years ago on Ang II dependent renal vascular tone of Notch3 -/- mice. The study is well tailored and adds insight into basic mechanisms of renal vascular tone regulation. 

Minor remark:

- please add information about mRNA expression of the genes listed in table 1 after administration of AngII

Author Response

We thank both reviewers for their comments that helped improving our manuscript. All changes are marked in red.

Minor remark:

- please add information about mRNA expression of the genes listed in table 1 after administration of AngII

Answer: The effect of Ang II is rapid and the vessel response is transient (15 sec to peak, return to baseline after 30-45 sec). In a such short period of time mRNA expressions of genes do not change. Only chronic administration of Ang II can alter genes expressions (for instance animals treated with minipumps for a long period of time), but these experiments address a different issue from the objectives of this present study.   

Reviewer 2 Report

In this manuscript, the authors examine the response of afferent arterioles to angiotensin II and KCl in notch3-/- mice. Although the role of notch3 in afferent arterioles is an interesting topic, the conclusions to the results in this study are questionable.

In notch3-/- afferent arterioles, EGTA, nifedipine, and miberadil showed exactly the same response to angiotensin II administration as normal medium. This result suggests that extracellular calcium influx does not occur in notch3-/- afferent arterioles in response to angiotensin II administration. The fact that extracellular calcium influx does not occur in both L-type channel inhibition and T-type channel inhibition suggests that this phenomenon is not caused by differences in channel types. On the other hand, notch3-/- afferent arterioles showed exactly the same reaction as wild-type to depolarization by KCL, suggesting that the L-type channels of notch3-/- afferent arterioles are intact. Considering these factors together, the decreased response of notch3-/- efferent arterioles to angiotensin II is not caused by the Ca channel itself, but is caused by an abnormality somewhere in the pathway from the angiotensin II receptor to the Ca channel.

The authors point to altered expression of mRNAs associated with t-type channels. However, the authors did not examine the protein-level expression or function of t-type channels, and there is no direct evidence of abnormalities in T-type channels in this manuscript.

Therefore, it is considered that the discussion and conclusions in this manuscript are not appropriate.

Author Response

We thank both reviewers for their comments that helped improving our manuscript. All changes are marked in red.

In this manuscript, the authors examine the response of afferent arterioles to angiotensin II and KCl in notch3-/- mice. Although the role of notch3 in afferent arterioles is an interesting topic, the conclusions to the results in this study are questionable.

General answer: We have modified several parts of the manuscript (abstract, discussion and conclusions) to address better the concerns of the reviewer. Below is our point by point answer to his/her remarks.

- In notch3-/- afferent arterioles, EGTA, nifedipine, and miberadil showed exactly the same response to angiotensin II administration as normal medium. This result suggests that extracellular calcium influx does not occur in notch3-/- afferent arterioles in response to angiotensin II administration.

Answer:  We agree with the reviewer.

- The fact that extracellular calcium influx does not occur in both L-type channel inhibition and T-type channel inhibition suggests that this phenomenon is not caused by differences in channel types.

Answer: The EGTA data show that calcium influx does not occur in Notch3-/- vessels. There are only L- and T-type channels in afferent arterioles. Thus, if Ca+2 entry is defective and since two types of channels exist, a logical hypothesis is to check whether these channels are expressed and/or functioning. 

- On the other hand, notch3-/- afferent arterioles showed exactly the same reaction as wild-type to depolarization by KCL, suggesting that the L-type channels of notch3-/- afferent arterioles are intact.

Answer: We agree with the reviewer.

- Considering these factors together, the decreased response of notch3-/- efferent arterioles to angiotensin II is not caused by the Ca channel itself, but is caused by an abnormality somewhere in the pathway from the angiotensin II receptor to the Ca channel.

Answer: The reviewer probably means afferent (and not efferent) arterioles. We agree that this could be an alternative explanation. However, our previous studies (Boulos et al, ref 6) demonstrated that the kidneys of Notch3-/- mice display deficient renal vascular responses to all tested vasoconstrictors such as angiotensin II, norepinephrine, thromboxane A2, and also to mechanical stress (no receptor signaling involved). Thus, the deficiency it is not specific to Ang II signaling. The common feature between these different ways of contraction is Ca+2

- The authors point to altered expression of mRNAs associated with t-type channels. However, the authors did not examine the protein-level expression or function of t-type channels, and there is no direct evidence of abnormalities in T-type channels in this manuscript.

Answer: The results with mibefradil show that Ca+2 does not enter through T type channels. Thus, either T-type channels are not expressed and/or are expressed but not functional. The data from differential analysis of mRNA show that the expression of T-type channels is not just altered, it is negligible (it is by far the most down regulated gene). In the absence of a detectable transcript, it is reasonable to expect that the protein expression will be negligible. Unfortunately, agents that allow to test functionality of T-type channels do not exist (such is the case with KCl and L-type channels), and we agree that the T-type expression is one among other hypotheses.

For this reason, we have modified the discussion and we propose other alternatives such as lack of maturation of VSMC, participation of the other genes involved in contractility (Cx43, transient receptor potential channel, rhoA) 

- Therefore, it is considered that the discussion and conclusions in this manuscript are not appropriate.

Answer: As said before, several parts of the text are modified in order to provide a thorough discussion of the data and the hypotheses.

Reviewer 3 Report

General comments

In this manuscript, the authors extended their previous study on the interaction between Notch3 and the contractility of renal resistance vessels. In this study, the authors have revealed that deletion on Notch3 impairs contractility of renal afferent arterioles possibly through the downregulation of functional expression of T-type Ca2+ channels. The key findings in this study are novel and interesting, however, this reviewer has several concerns as stated below.

 Specific comments

1.      It is a pity that the authors did not investigate the role of T-type Ca2+ channels in Ang II-induced vasoconstriction in renal efferent arterioles using Notch3-/- mice. Because the Ang II-induced vasoconstriction is predominantly dependent on T-type Ca2+ channels in efferent arterioles, the authors might have been able to better clarify the interaction between Notch3 and T-type Ca2+ channels using efferent arterioles. Why did not the authors use efferent arterioles in the present study?

2.      While the authors speculated that the L- and T-type channel blockers have no additional effects on dilation by citing a previous study in rat juxtaglomerular afferent arterioles, this hypothesis was not tested in the present study. Because the vasoreactivity greatly varies not only with the type of vessel studied but also with the species, the authors need to test this hypothesis in renal afferent arterioles of Notch3-/- mice.

3.      It seems odd that nifedipine did not inhibit the Ang II-induced vasoconstriction in afferent arterioles of Notch3-/- mice (Fig 3C) considering the fact that the function of L-type Ca2+ entry per se appears to be intact in this vessel (Fig 5A, B). Are there any possibility that AT2 receptor-mediated vasorelaxant effects of Ang II counteract the vasoconstricting effects of Ang II in the afferent arterioles of Notch3-/- mice?

4.      It seems strange to this reviewer that nifedipine alone completely inhibit the high KCl-induced vasoconstriction in the afferent arterioles of wild type mice (Fig 5A, B). Does this mean T-type Ca2+ channels cannot be activated by depolarizing stimuli in this vessel? If so, what are the underlying mechanisms of mibefradil-sensitive vasoconstriction in the afferent arterioles of wild type mice constricted with 10-7 M Angiotensin II (Fig 3A, D) ?

Author Response

Reviewer 3

Specific comments

1. It is a pity that the authors did not investigate the role of T-type Ca2+channels in Ang II-induced vasoconstriction in renal efferent arterioles using Notch3-/-mice. Because the Ang II-induced vasoconstriction is predominantly dependent on T-type Ca2+ channels in efferent arterioles, the authors might have been able to better clarify the interaction between Notch3 and T-type Ca2+ channels using efferent arterioles. Why did not the authors use efferent arterioles in the present study?

Answer: We agree that exploring T-type channels in efferent arterioles could be an interesting subject. However, in the particular case of our study with Notch3-/- mice, the objective was to understand the deficient myogenic response in the kidney that we have observed previously (ref 6). Myogenic response in the kidney depends on afferent arterioles and not on efferent. A general study on the role of T-type channels in efferent arterioles and the comparison with afferents is far beyond the scope of the present study.

2. While the authors speculated that the L- and T-type channel blockers have no additional effects on dilation by citing a previous study in rat juxtaglomerular afferent arterioles, this hypothesis was not tested in the present study. Because the vasoreactivity greatly varies not only with the type of vessel studied but also with the species, the authors need to test this hypothesis in renal afferent arterioles of Notch3-/-mice.

Answer: In Figure 3, Notch3-/- vessels under control conditions (A) display a deficient response which  is the same with that observed with the complete absence of Ca+2 from the medium (EGTA, panel B). This means that there is no entry at all of Ca+2 in these vessels. The response to blockers Nifedipine (Fig 3 C) or mibefradil (Fig 3 D) is similar to EGTA, meaning that Ca+2 is not entering in the Notch3-/- vessels. In addition, Thapsigarine (Figure 4) shows that the Ca+2 response of Notch3-/- is due to intracellular Ca+2 mobilization. Thus, it is not excpected to have any additive effect of the blockers, since there is no entry to begin with.

3. It seems odd that nifedipine did not inhibit the Ang II-induced vasoconstriction in afferent arterioles of Notch3-/-mice (Fig 3C) considering the fact that the function of L-type Ca2+ entry per se appears to be intact in this vessel (Fig 5A, B). Are there any possibility that AT2 receptor-mediated vasorelaxant effects of Ang II counteract the vasoconstricting effects of Ang II in the afferent arterioles of Notch3-/- mice?

Answer: In our previous study (Boulos et al, ref 6) we have shown that the kidneys of Notch3-/- mice display deficient renal vascular responses to all tested vasoconstrictors such as angiotensin II, norepinephrine, thromboxane A2, to mechanical stress (no receptor signaling involved) and to vasodilators as well (prostacyclin-cAMP pathway, bradykinin-cGMP pathway). Thus, it is unlikely that the deficient vasoconstriction is due to a counteracting vasodilatory mechanism. We have added this precision to the discussion (page 7).  

4a. It seems strange to this reviewer that nifedipine alone completely inhibit the high KCl-induced vasoconstriction in the afferent arterioles of wild type mice (Fig 5A, B). Does this mean T-type Ca2+ channels cannot be activated by depolarizing stimuli in this vessel?

Answer: This is not a new finding. Before us, several other investigators have reported that nifedipine alone completely inhibits KCl-induced vasoconstriction in afferent arterioles (ref 22-25). Some proposed that the function of T-type channels has little to negligible contribution, others believe that T-type channels are functional but independent to constrictor responses to KCl. It appears likely that the low-current transient T-type calcium flow happens, but is unable to activate the high-conductance L-type channels because its mobilization is not enough to displace the nifedipine blockade. As result, the KCl-induced vasoconstriction is completely inhibited in presence of nifedipine.

This point is now discussed in page 7.

4b. If so, what are the underlying mechanisms of mibefradil-sensitive vasoconstriction in the afferent arterioles of wild type mice constricted with 10-7 M Angiotensin II (Fig 3A, D) ?

Answer: The T-type channel firing is rapid, very short and transient (in less than 40 ms the T-channel calcium flux is back to zero) (Rossier M, Front Endocrinol (Lausanne). 2016;7:43). The inhibition of the T-type channels with mibefradil breaks the cascade of Ca+2 before the L-type channels. Ca+2 must be augmented by the transient T-type flux, to thereafter activate the L-type channels which require a much higher increase in voltage. If the T-type is inhibited, then the initial steps of this interaction are broken affecting the subsequent L-type Ca+2 entry.

This point is now discussed in page 7.

Round 2

Reviewer 2 Report

In this manuscript,

1. In Notch3-/- afferent arterioles, the influx of extracellular calcium does not occur in response to the administration of angiotensin II.

2. Changes in the expression of mRNAs related to t-type channels,

3. Against depolarization by KCL, notch-/- afferent arterioles showed exactly the same reaction as the wild type.

These three points are the results. What the authors described in the discussion is a hypothesis, and it is not possible to conclude from the results shown in this manuscript that “Deletion of Notch3 impairs contractility of renal resistance 2 vessels through a T-type Ca2+ channel-dependent mechanism”. Therefore, the title and conclusions of this manuscript are not appropriate.

Author Response

Reviewer 2

In this manuscript,

  1. In Notch3-/- afferent arterioles, the influx of extracellular calcium does not occur in response to the administration of angiotensin II.
  2. Changes in the expression of mRNAs related to t-type channels,
  3. Against depolarization by KCL, notch-/- afferent arterioles showed exactly the same reaction as the wild type.

These three points are the results. What the authors described in the discussion is a hypothesis, and it is not possible to conclude from the results shown in this manuscript that “Deletion of Notch3 impairs contractility of renal resistance 2 vessels through a T-type Ca2+ channel-dependent mechanism”. Therefore, the title and conclusions of this manuscript are not appropriate.

Answer: To comply with the reviewer's request we have :

- modified the title to “Deletion of Notch3 impairs contractility of renal resistance 2 vessels due to deficient Ca2+ entry”.

- removed from the conclusions the sentence "This absence of a sustained Ca2+ response was caused by compromised expression and function of the T-type Ca2+ channel, an established regulator of cell excitability."

- remodeled the first sentence of the conclusions to "The present study is to our knowledge the first to identify deficient Ca2+ entry as a major vasoactive signaling pathway dependent on Notch3 for proper function, which resulted in a serious defect of the calcium handling in renal resistance vessels of mice lacking Notch3-/- expression."

Reviewer 3 Report

The authors have addressed all the concerns raised by this reviewer. Could the authors please explain why efferent arterioles were not used in the present study in the introduction or in method section?

Author Response

Thank you the reviewer for his/her comments.

Following his/her suggestion a comment is now inserted in the methods section explaining the reason to not use efferent arterioles in this study

Round 3

Reviewer 2 Report

The revised manuscript has been improved in response to the reviewer's comments. This time, there are no particular problems to point out.

Author Response

Thank you the reviewer for helping  to improve our manuscript